# Non-inferiority acceptance testing for a CT body composition algorithm

**Daniel J. Blezek**[1]      BLEZEK.DANIEL@MAYO.EDU
**Michael Moynagh**[1]      MOYNAGH.MICHAEL@MAYO.EDU
**Hillary Garner**[2]      GARNER.HILLARY@MAYO.EDU
**Patrick Navin**[1]      NAVIN.PATRICK@MAYO.EDU
**Doris Wenger**[1]      WENGER.DORIS@MAYO.EDU

[1] *Mayo Clinic, Radiology, 200 First Street SW, Rochester, MN, 55905, USA*
[2] *Mayo Clinic, Radiology, 4500 San Pablo Rd S, Jacksonville, FL 32224, USA*

**Editors:** Under Review for MIDL 2021

## Abstract

Value of an algorithm in a clinical setting is difficult to gauge. Our hypothesis for this study is that the acceptance rate of an automated segmentation algorithm for determination of body composition is non-inferior to manual segmentation. A panel of four abdominal radiologists reviewed blinded segmentation results from human raters and the algorithm and were asked to accept or reject results based on the question **"I am confident these results are reasonably accurate, and would record the measurements into the patient record"**. Segmentations were accepted at a rate of 82% overall, 82% for the algorithm and 84% for manual and the algorithm was non-inferior to manual segmentation with $p < 0.025$, however, the study did not reach 80% power. We conclude the algorithm performs as well as manual segmentation but a larger study is required for proper power.

**Keywords:** body composition, non-inferiority testing, CT, segmentation

## 1. Introduction and Background

Gauging the clinical impact of an AI algorithm is difficult. Many factors influence the clinical utility of an algorithm including predictive value, workflow integration, quality of results, interpretability, etc. Body composition, the distribution of adipose tissue and skeletal muscle, has been linked outcomes in several clinical conditions, yet the potential for clinical impact is as yet untapped. This highlights the need for a simple, automated algorithm that is easy to verify and report in the workflow of busy radiologists.

The body composition AI algorithm identifies subcutaneous adipose tissue (SAT), visceral adipose tissue (VAT), and skeletal muscle (SKM) from a slice at roughly the S/I center of the L3 vertebral body from an abdominal CT study (Weston et al., 2019). The algorithm operates completely automatically and has been trained on over 2,000 example images. Despite the large training dataset, anatomic variability leads to unavoidable errors in the algorithm. Here we propose a simple experiment to determine it's suitability for clinical use. The experiment is for radiologists to assess algorithm segmentation results to manual segmentation by 3d lab technologists using FDA approved software. The rational for this work is that, currently, to implement a body composition service in a radiology department would require manual segmentation by FDA approved software as no FDA approved AI

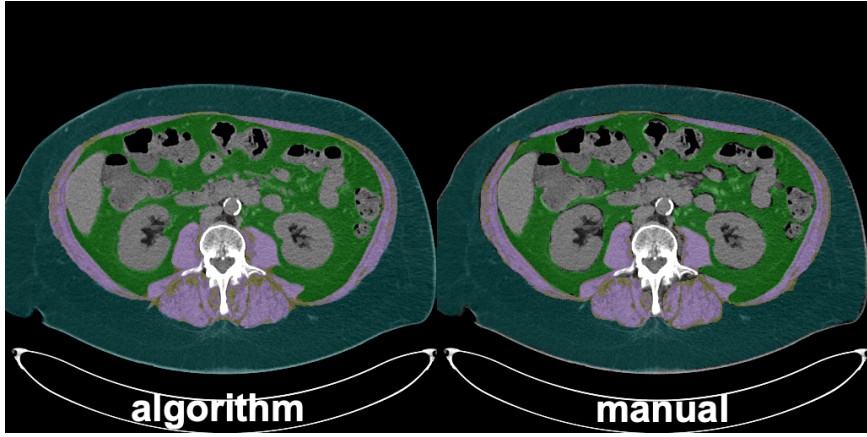

Figure 1: Comparison of algorithm and manual segmentation of subcutaneous adipose tissue (light green), visceral adipose tissue (purple), skeletal muscle (green)

algorithms exist. However, the time required for each case prohibits a manual approach. To be useful, an algorithm must be shown to be non-inferior to manual segmentation to some clinically acceptable degree (Liu et al., 2002). Rather than evaluate similarity scores, in this study, we hypothesize that the "acceptance rate" of an automated, ML algorithm for body composition is non-inferior to manual segmentation in a clinical context, *i.e.* how often would a radiologist accept and report the result.

## 2. Methods

To test our hypothesis, we conducted a simple experiment. Four radiologists (HG, MM, PN, DW) were asked to rate single-slice segmentations. In one group, experienced 3D lab technologists identified regions in the slice (the "manual" group), and in the second group, the AI algorithm segmented the same slice (the "algorithm" group). The radiologists were blinded to the two groups and evaluated over two reading sessions. Each reading session consisted of a random choice of the manual or algorithm result for each subject. Thus each radiologist rated all patients twice, once with manual segmentation and once with algorithm segmentation in randomized batches in two reading sessions separated by an appropriate "forgetting" period.

CT abdomen pelvis images (N=49) were selected with proper IRB approval. Experienced 3d lab technologists were instructed to select a single slice from the S/I center of the L3 vertebral body and segment 3 structures: SAT, VAT, and SKM using the TeraRecon software (TeraRecon, Inc. Durham, NC 27703 USA). The same slice was presented to the algorithm for segmentation. The segmentations were converted to DICOM images using custom software for consistent presentation, see Figure 1. Radiologists were instructed to approve or reject segmentations where approval is a positive response to the question **"I am confident these results are reasonably accurate, and would record the measurements into the patient record"**.

## 3. Results

Table 1: Acceptance rates by rater expressed as accepted / reject and rates

|         | Overall          | Algorithm        | Manual           |
|---------|------------------|------------------|------------------|
| Rater A | 80 / 18 (0.82)   | 36 / 13 (0.73)   | 44 / 5 (0.90)    |
| Rater B | 72 / 26 (0.73)   | 35 / 14 (0.71)   | 37 / 12 (0.76)   |
| Rater C | 82 / 16 (0.84)   | 43 / 6 (0.88)    | 39 / 10 (0.80)   |
| Rater D | 91 / 7 (0.93)    | 47 / 2 (0.96)    | 44 / 5 (0.90)    |
| **Total** | 325 / 67 (0.82) | 161 / 35 (0.82)  | 164 / 32 (0.84)  |

The results of this study are summarized in Table 1. The CT cases were retrieved from the imaging archive and manually segmented by the 3d technologists. The same slice was processed by the algorithm. Both segmentations were converted into DICOM and sent to a PACS workstation. Of all the results, 82% were rated as acceptable, 82% for the algorithm and 84% for the manual segmentation. Raters "C" and "D" slightly favored the algorithm, while "A" and "B" favored manual segmentations. The algorithm was non-inferior to acceptance of manual segmentation with $p < 0.025$ for a difference in acceptance rate $\delta < 0.1$. The study was powered at 54% for non-inferiority of paired data for 196 observations of 49 subjects (Liu et al., 2002). 396 paired observations are required for 80% power with expected acceptance rate of 0.8, 2.5% one-sided type I error.

## 4. Discussion and Conclusion

Our small study indicates an automated algorithm for identification of SAT, VAT and SKM would be accepted at the same rate as manual segmentation. This experiment is in agreement with prior objective comparisons (Weston et al., 2019) and establishes initial evidence of acceptance rates of roughly 80% that can be used in power calculations for an expanded trial to establish non-inferiority of the algorithm.

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
