# OpenReview forum: "Non-inferiority acceptance testing for a CT body composition algorithm"
_MIDL.io/2021/Conference/Short — Submitted to MIDL 2021_

### Official Review · Reviewer_PLKp · 2021-04-29

**Confidence:** 4
**Final Rating:** 2

**Summary:**

The short paper solely describes the evaluation of an algorithm that is not discussed itself. The evaluation targets the question whether the algorithm performs a simple and relevant semantic segmentation problem with three foreground tissue classes not (significantly) inferior to a manual segmentation based on ratings by four radiologists. The results (based on 49 subjects) are not very conclusive; two radiologists favored the algorithm results and two favored manual segmentations, and the threshold for non-inferiority of the acceptance rate was set to 10%.

**Strengths:**

The body composition assessment task addressed by the automatic algorithm is clinically relevant, and the question whether the algorithm is non-inferior to manual segmentation (which is not feasible in clinical practice) is valid.
The authors have applied proper statistical techniques to this question, and the presentation is clear enough to understand exactly what has been done.

**Weaknesses:**

Non-inferiority was defined to a margin of 10% lower acceptance rate, for which no justification is given. This number is not particularly low. The authors state that "to be useful, an algorithm must be shown to be non-inferior to manual segmentation to some clinically acceptable degree", but no rationale is given for why 10% difference is "clinically acceptable".
The paper itself states a major weakness, namely that the data analyzed was not sufficient; citing the conclusion: "The study was powered at 54% for non-inferiority of paired data…" which means that the only question the paper addresses could not be reliably answered. The authors are very honest about this, but that means that the contribution is vanishingly small.

**Deanonymize Review:**

yes

**Detailed Comments:**

The choice of colors in Fig. 1 is ok, but the caption gives the wrong association of colors and tissue classes.

Spelling: "it's", "rational[e]"

**Justification Of The Rating:**

The contribution is very minor, as described under "weaknesses". As the authors conclude, the main contribution is that a proper power calculation for a larger study becomes possible. The algorithm itself is not described at all.

**Paper Type:**

validation/application paper

**Special Issue:**

no

---

### Official Review · Reviewer_fGnH · 2021-05-07

**Confidence:** 5
**Final Rating:** 1

**Summary:**

The paper presents an evaluation of automatic segmentation vs human rater manual segmentation by having segmentations rated by radiologists as reasonably accurate and appropriate for patient records. The evaluation shows that overall 82% of the automatic segmentation were judged as appropriate vs 84% for the manual segmentations.

**Strengths:**

- Radiologist evaluation based assessment of automatic segmentation, judges the segmentation as whether appropriate for recording in patient records
- 4 radiologists were employed
- intermediate size data (N=49)

**Weaknesses:**

Major:
- single slice segmentation, inappropriate for a 3D structure
- overall data sample is far too small
- the authors looked group differences between acceptance of the 2 segmentations (automatic vs manual) and when a given p-value was not reached, then they concluded that there is "The algorithm was non-inferior to acceptance of manual segmentation with p < 0.025". That is a statistically incorrect statement. The statistical testing here tests whether the hypothesis that the 2 measurement distributions are same (H0) can be safely rejected at a given p-value threshold (commonly 5% or 1%). Apart from the unusual 0.025 p-value threshold (not sure why the actually observed p-value is not reported), this does NOT make a statement about the sameness of the measurements when the p-value is not reached. Thus, one cannot conclude that the 2 segmentations (automatic vs manual) are the same (or one is non-inferior to the other) because a 0.025 p-value threshold was not reached, but rather one can only conclude that there is insufficient evidence that they are different given the current (quite small) dataset.
- no presentation of the automatic deep learning algorithm (basically only a citation of the method)

Minor: radiologists are not necessarily the best in assessing segmentation accuracy, other medical professions that deal in the quantification of anatomical structures may be sometimes more appropriate (e.g. oncologists, neuroanatomists etc), undiscussed

**Deanonymize Review:**

no

**Justification Of The Rating:**

- Too small study on a single slice with insufficient evidence to derive meaningful knowlegde
- Incorrect interpretation of the observed statistical results
- needs reporting of actually observed p-values as well as at least a single sentence on how the segmentation method works

**Paper Type:**

validation/application paper

**Special Issue:**

no

---

### Meta-Review · Program_Chairs · 2021-05-10

**Recommendation:** Reject
**Confidence:** 5

**Metareview:**

The contribution of the paper is very small, results are inconclusive, and there are concerns about the validity of the statistical tests performed.

---

### Decision · Program_Chairs · 2021-05-11

Reject